# HGPRT and PNP: Recombinant Enzymes from *Schistosoma mansoni* and Their Role in Immunotherapy during Experimental Murine Schistosomiasis

**DOI:** 10.3390/pathogens12040527

**Published:** 2023-03-29

**Authors:** Bruna Dias de Lima Fragelli, Ana Carolina Maragno Fattori, Elisandra de Almeida Montija, Joice Margareth de Almeida Rodolpho, Cynthia Aparecida de Castro, Krissia Franco de Godoy, Camila Tita Nogueira, Vanderlei Rodrigues, Edson Garcia Soares, Larissa Romanello, Juliana R. Torini, Humberto D’Muniz Pereira, Fernanda de Freitas Anibal

**Affiliations:** 1Laboratório de Inflamação e Doenças Infecciosas, Departamento de Morfologia e Patologia, Universidade Federal de São Carlos, São Carlos 13565-905, Brazil; 2Departamento de Bioquímica, Escola Paulista de Medicina, Universidade Federal de São Paulo, São Paulo 04039-032, Brazil; 3Departamento de Bioquímica e Imunologia, Faculdade de Medicina, Universidade de São Paulo, Ribeirão Preto 14040-900, Brazil; 4Laboratório de Citopatologia, Departamento de Patologia e Medicina Legal, Universidade de São Paulo, Ribeirão Preto 14040-900, Brazil; 5Departamento de Saúde e Psicologia, Universidade do Estado de Minas Gerais, Unidade Ituiutaba, Ituiutaba 38302-192, Brazil; 6Laboratório de Biologia Estrutural, Instituto de Física de São Carlos, Universidade de São Paulo, São Carlos 13565-905, Brazil

**Keywords:** schistosomiasis, *Schistosoma mansoni*, immunotherapy, purine salvage pathway, HGPRT, PNP, granuloma, inflammation

## Abstract

Schistosomiasis is a parasitic infection caused by trematode worms (also called blood flukes) of the genus *Schistosoma* sp., which affects over 230 million people worldwide, causing 200,000 deaths annually. There is no vaccine or new drugs available, which represents a worrying aspect, since there is loss of sensitivity of the parasite to the medication recommended by the World Health Organization, Praziquantel. The present study evaluated the effects of the recombinant enzymes of *S. mansoni* Hypoxanthine-Guanine Phosphoribosyltransferase (HGPRT), Purine Nucleoside Phosphorylase (PNP) and the MIX of both enzymes in the immunotherapy of schistosomiasis in murine model. These enzymes are part of the purine salvage pathway, the only metabolic pathway present in the parasite for this purpose, being essential for the synthesis of DNA and RNA. Female mice of Swiss and BALB/c strains were infected with cercariae and treated, intraperitoneally, with three doses of 100 µg of enzymes. After the immunotherapy, the eggs and adult worms were counted in the feces; the number of eosinophils from the fluid in the peritoneal cavity and peripheral blood was observed; and the quantification of the cytokine IL-4 and the production of antibodies IgE was analyzed. The evaluation of the number of granulomas and collagen deposition via histological slides of the liver was performed. The results demonstrate that immunotherapy with the enzyme HGPRT seems to stimulate the production of IL-4 and promoted a significant reduction of granulomas in the liver in treated animals. The treatment with the enzyme PNP and the MIX was able to reduce the number of worms in the liver and in the mesenteric vessels of the intestine, to reduce the number of eggs in the feces and to negatively modulate the number of eosinophils. Therefore, immunotherapy with the recombinant enzymes of *S. mansoni* HGPRT and PNP might contribute to the control and reduction of the pathophysiological aspects of schistosomiasis, helping to decrease the morbidity associated with the infection in murine model.

## 1. Introduction

Schistosomiasis is a disease caused by intravascular worms of the genus *Schistosoma* sp., and the species *Schistosoma mansoni*, endemic in Africa, the Antilles and South America, causes one of the schistosomiasis affecting the digestive tract [1,2,3]. According to estimates, approximately 251.4 million people required preventive treatment for schistosomiasis in 2021, and more than 75.3 2 million people were reported to have been treated [4].

The infection occurs through contact with water contaminated with penetrating parasite larvae during agricultural, domestic, and recreational activities in tropical and subtropical countries, where there are communities without access to drinking water and with inadequate basic sanitation, with reduced investments in research, production of medicines and control of this illness, making schistosomiasis a neglected disease. Schistosomiasis is a recurring focus in health debates around the world, as it is global in scope and is the second in the ranking of tropical diseases in mortality [1,2,4].

The disease control protocol is based on periodic large-scale treatment of the population with the only drug recommended by WHO, Praziquantel (PQZ), which is effective against all six species of *Schistosoma* sp. There is also a second, more comprehensive approach, which includes water and sewage treatment and disease vector control of snails of the genus *Biomphalaria* ssp. [3].

There are studies that describe the loss of sensitivity of the parasite to PZQ. These studies show worms isolated from patients from three hospitals in different countries who showed a need for a dose much higher than the ED_50_ (dose needed to kill 50%), showing the urgent need to investigate new drugs, vaccines and alternative treatments for schistosomiasis [5,6]. PZQ is also not effective against schistosomula and *S. mansoni* eggs [7].

*S. mansoni*, unlike its human host, depends solely and completely on the purine salvage pathway to supply its demands for nucleotides [8,9,10]. The main enzymes of the purine salvage pathway in *S. mansoni* are phosphoribosyltransferases, which catalyzes the conversion of purine bases hypoxanthine and guanine to their respective nucleotides IMP and GMP in the presence of 5-phosphorylibose 1-pyrophosphate (PRPP). The Purine Nucleoside Phosphorylase (PNP) is an enzyme responsible for the reversibility of phospholysis of purine nucleotides, generating ribose-1-phosphate and its corresponding bases [11]. HGPRT catalyzes the removal of the pyrophosphate group (PPi) from phosphoribosyl-pyrophosphate (PRPP) and the addition of a purine base (hypoxanthine or guanine), with the cofactor Mg^2+^. This reversible reaction’s end result is the formation of IMP (inosine monophosphate) or GMP (guanine monophosphate), which favors the formation of nucleotides [12].

These enzymes have been shown to be potential targets for therapy and immunization against schistosomiasis [8,13], since their inhibition effectively blocks the supply of guanine nucleotides [14]. 

Therefore, the present study aimed to verify the role of the recombinant enzymes of *S. mansoni*, HGPRT and PNP, during experimental schistosomiasis in mice in the phase of early maturity of adult worms and egg laying, since the drug PZQ does not affect these stages of development of the parasite, as a possible immunotherapy candidate for patients in endemic areas.

## 2. Materials and Methods

### 2.1. Obtaining the S. mansoni HGPRT and PNP Enzymes

The *S. mansoni* genome project database, HGPRTs, specifically Smp_103560.1, was searched to find the coding sequence for the hypoxanthine-guanine phosphoribosyltransferase isoforms. In the *S. mansoni* genome, the SmPNP2 gene (Smp_179110) has been discovered. Recombinant enzymes were expressed in system using *E. coli* BL21 (ED3) in sufficient quantity for the experiments. The enzymes were purified by the affinity chromatography method as previously described [11,15], in collaboration with the Structural Biology Laboratory of the Center for Molecular Biology Structural (CBME) of the Institute of Physics of São Carlos (IFSC), located at the University of São Paulo (USP), São Carlos, Brazil, coordinated by Dr. Humberto d’Muniz Pereira.

### 2.2. Animals

Female mice of Swiss and BALB/c strains, weighing between 18–20 g and between 4–5 weeks of age, from the General Bioterium Hospital of the City Hall of the University of São Paulo (USP) (Ribeirão Preto, Brazil) were used for the assay. These animals have the Specified Pathogen Free (SPF) certificate. All animals were kept in the bioterium of the Morphology and Pathology Department of the Federal University of São Carlos (DMP—UFSCar) (Brazil), with free access to water and food for rodents. 

The experimental design of this study was based on the recommendations of the Ethical Principles of Animal Experimentation adopted by the Brazilian Society of Science of Laboratory Animals and approved by the Ethics Committee on Animal Use of the Federal University of São Carlos, under protocol number 7939170816. 

### 2.3. Obtaining the Infectious Larvae of S. mansoni and Animal Infection

Cercariae, larval stage capable of infecting, of the LE strain were kindly provided by Prof. Dr. Vanderlei Rodrigues, from the Department of Biochemistry and Immunology, Faculty of Medicine of the University of São Paulo (USP) (Brazil), which were released from infected snails of the genus *Biomphalaria*. The received the cercariae by the subcutaneous route with 80 larva/0.5 mL of saline/animal were inoculated. 

### 2.4. Animal Immunotherapy

Immunotherapy was performed in three doses on the 28th, 38th and 48th day after infection. Each dose contained 100 µg of each enzyme that have been emulsified with 100 µg of aluminum hydroxide, Al(OH)_3_, diluted in PBS 1X (Phosphate Buffered Saline: 8 g NaCl, 0.2 g KCl, 1.15 g Na_2_HPO_4_, 0.2 g KH_2_PO_4_ in 1 L of ultra-pure water), totaling 100 µL/animal intraperitoneal route. The dose was established by the standardization carried out by the research group [16,17] The experimental groups were divided (Table 1). 

Table 1 Groups and therapy. Six experimental groups were created: Control, in which the animals received neither infection nor treatment; Sm, in which the animals received only infection; HGPRT, in which the animals received only treatment with the recombinant HGPRT enzyme; Sm + HGPRT, in which the animals received infection and treatment with the recombinant HGPRT enzyme; Sm + PNP, in which the animals received infection and treatment with the recombinant enzyme PNP; and Sm + MIX, in which the animals received infection and treatment with both recombinant enzymes.

The study using only HGPRT as a treatment was previously carried out in our laboratory. As the results proved to be very promising, a new project was established with one more enzyme from the purine salvage pathway, PNP, now using the Balb/c strain. The proposed model (Figure 1) was repeated twice to confirm the results obtained.

### 2.5. Analysis of Parasitological Data

The amount of eggs disposed of in the feces individually from infected and treated animals was analyzed with the Kato–Katz method (Katz and Peixoto, 2000) on the 54th days after infection. The 54-day trials were performed to accurately assess the impact of the recombinant enzyme on egg laying and in the liver. By passing a predetermined volume of feces through an aperture with a given diameter, the feces were sieved in a filter test (Helm-Test) and displayed on microscopic slides. To preserve excrement and bleach *S. mansoni* eggs, the slides were covered with a cellophane coverslip coated with malachite green. After reading the slides and counting the eggs, the number of eggs per gram of feces was calculated using the following formula: Eggs per gram of feces = amount of eggs in the slide × 24 (orifice correction factor Helm − Test)

Adult worms were examined, taken out of the intestinal mesentery, and a procedure called perfusion of the hepatic porta vein was done to count how many there were [18]. The decrease in the number of worms was measured by comparing the number of worms recovered in each experimental group and their respective control (Sm group), according to the formula:% LPB = (NWC − NWE/NWC) × 100
where % LPB is the lowering the parasitic burden, NWC is the number of worms recovered from the control group and NWE is the number of worms recovered from the experimental group [19].

### 2.6. Cell Evaluation

On the 55th day after *S. mansoni* infection, the collection of peripheral blood (PB) was performed by puncture of the left brachial vein, with syringes bathed with anticoagulant EDTA 0.3 M. For the collection of the liquid from peritoneal cavity (LPC), PBS 1X containing 0.5% sodium citrate (PBS/Citrate) was used, with 3.0 mL applied to the animals’ peritoneum. Subsequently, after homogenizing the area, the peritoneal cells were retrieved. The count of total leukocytes of PB and LPC was performed individually in the Neubauer chamber, in which each sample was diluted in Turk solution (15 mL glacial acetic acid, 1 mL 2% gentian violet, 500 mL milli-q water) in the proportion 1:20. For the differential leukocyte count, LPC slides were prepared in the cytocentrifuge (1500 g/3 min) and blood smear, both stained with the Panotic–Laborclin dye.

### 2.7. Dosage of Cytokine IL-4 and Detection of Antibody IgE by the ELISA Method

The animals’ plasma was obtained after centrifugation of the whole blood in 1500 g/15 min. Antibodies and cytokines were investigated by the ELISA immunoenzymatic assay, following the manufacturer’s instructions, for IgE and IL-4 (OptEIA ™ Kit, BD Biosciences), briefly described: in 96 microtiter plates wells (Corning^®^ Costar 3590), 8 µg/well of enzymes (HGPRT or PNP) diluted in 0.1 M carbonate buffer pH 9.5 were applied for sensitization to IgE antibody, and the plates were incubated for 16 h at 4 °C. Protein extraction was performed using the extraction buffer (Tris-HCl, DTT and Glycerol). The sample (adult worms) was treated with a protease inhibitor (Sigma-ALDRICH, St. Louis, MI, USA), sonicated and centrifuged, and the pellet discarded. Subsequently, the quantification of the protein extract (supernatant) was performed by the Lowry method (LOWRY et al., 1951). A 15% polyacrylamide gel as made using the SDS-Page technique to verify the purity of the sample. Sensitization for the cytokine IL-4 was carried out for 16 h at 4 °C using the corresponding primary monoclonal antibody diluted in 0.1 M carbonate buffer pH 9.5 in a final volume of 100 µL/well. The plates were washed with 300 µL/well of PBS 0.05% Tween 20, pH 7.4, after sensitization. (washing solution). The plates were washed, 200 µL/well of blocking solution (PBS + 1% BSA) was added, and the incubation period was 1 h. The plates were then washed once more after that. Then, the standard protein (cytokine kit) and samples were applied to the plates, with 100 μL/well of the plasma pool of mice from each experimental group, applied and the plates were incubated for 2 h. The samples were diluted in 1:10 carbonate buffer. After that, the plates were washed and the secondary antibody conjugated to the peroxidase enzyme was added, diluted in PBS + 1% BSA, in different proportions for each antibody and cytokine, according to the manufacturer, and 100 µL/well were added. The plates were then incubated for 1 h and 30 min protected from light. Then, the plates were washed; 100 µL/well of TMB substrate (3.3′, 5.5′—Tetramethylbenzidine—BD Bioscience) was applied; and the plates were incubated, protected from light, for up to 30 min. Application of 50 L/well of 2 M sulfuric acid stopped the process (stop solution). The plates were read at a wavelength of 450 nm in an ELISA plate reader (Thermo Scientific ™ Multiskan ™ GO Microplate Spectrophotometer, Thermo Scientific, Vantaa, Finland).

### 2.8. Histology of the Liver

The liver of two mice per experimental group was removed on the 55th day after infection. The organs were washed with PBS 1X and carefully dried on absorbent paper. They were then fixed in buffered formaldehyde (4 g NaH_2_PO_4_, 6.5 g Na_2_HPO_4_, 100 mL formaldehyde and 900 mL distilled water). The histological slides were produced with fragments of each organ included in paraffin blocks and sectioned in 4 μm sections. Hematoxylin–Eosin (HE) was used to stain the slides, to observe tissue cells and granulomas, and Gomori’s Trichomic (GT) to analyze collagen deposition. The data are shown qualitatively.

The slides were read by counting eggs and granulomas present in the histological sections of the liver; they were scanned at Pannoramic Desk, 3D Histech’s Ltd., Hungary a multi-user equipment located at the Applied Immunology Laboratory, in the Department of Genetics and Evolution (DGE), from the Federal University of São Carlos (UFSCar), under the responsibility of Prof. Ricardo Carneiro Borra. The images were produced using the Pannoramic Viewer number 1.15.4 3D Histech’s Ltd., Hungary.

### 2.9. Statistical Analysis

GraphPad Prism 7.0 was used to analyze the data gathered for this investigation (San Diego, CA, USA). The study involved two separate experiments. The Grubbs analysis was used to find the inconsistent data, and the Shapiro-Wilk test was used to determine whether the data were parametric or non-parametric. As a result, the parametric data were subjected to the ANOVA test (analysis of variance) and the Tukey multiple comparisons post-test (results were presented in mean and standard deviation). The Kruskal-Wallis test and the Dunn multiple comparison post-test were employed for nonparametric data. (results were presented as the median with the upper and lower quartiles: Me [Q1; Q3]). The *t* test was used to compare the two groups, and non-parametric.

## 3. Results

Figure 1 shows the results of the immunotherapy applied in female Swiss mice. There was a significant reduction of 69.5% in the number of eggs in the feces (Figure 1A) and 30.06% of recovered adult worms in animals treated with HGPRT (Figure 1B). Figure 1C shows the absolute values of worms and is divided into couples, females and males. The number of eosinophils in the LPC and in the blood is shown in Figure 1D,E, in which the number of these cells is greater in infected animals and there is a decrease in animals treated with HGPRT when compared to infected and untreated animals.

The next figures show the results obtained in BALB/c mice. Figure 2A shows the significant decrease in the number of eggs in the feces of 45.6%, 59% and 27.5% in animals treated with HGPRT, PNP and MIX, respectively. The reduction in the number of recovered adult worms is shown in Figure 2B,C where a reduction of 3.6%, 12.6% and 11.6% is noted in animals treated with HGPRT, PNP and MIX, respectively. The Figure 2D,E show the eosinophil count in the LPC and blood. There is a significant increase in eosinophils obtained from Sm and Sm + HGPRT and Sm + MIX groups when compared to the control group in LPC. The same response profile obtained in the LPC was observed in the blood, but with no significant difference.

Figure 3A shows the cytokine levels of all groups. The IL-4 cytokine showed higher levels in the Sm + HGPRT group. Figure 3B–E reflect the averages and standard deviations (SD) from the mice plasma pool from the experimental groups coated with IgE antibody detection *S. mansoni* total proteins—STP (B), HGPRT enzyme (C), PNP enzyme (D), MIX enzymes (E). The therapy with the recombinant enzymes of *S. mansoni* was able to induce the synthesis of IgE antibodies when the well was coated with the enzymes HGPRT, PNP, MIX, and STP.

Figure 4 shows the histological analysis of the animals’ liver. Figure 4A represents histological sections stained with HE, showing the formation of granulomas. In Figure 4B, the liver slices were stained with GT to assess collagen deposition. It is feasible to see that the control group’s animals’ livers exhibit preserved tissue and without the presence of granulomas or eggs and, in the animals of the Sm, Sm + HGPRT, Sm + PNP and Sm + MIX groups, there is the formation of periovular granuloma, with mixed cellular infiltrate and fibrosis of the vessels. In Sm group, there is a mixed inflammatory infiltrate with dense inflammation, infiltrates of lymphocytes, macrophages and giant cells, live eggs and degenerated larvae. In the Sm + MIX group, the granulomas are well structured, and the eggs are preserved. It is also possible to observe the minimum portal collagen deposit (in blue) in the liver of the animals in the control group and in the animals in the Sm, Sm + HGPRT, Sm + PNP and Sm + MIX groups (the development of necrosis, the presence of a collagen layer surrounding the ves-sels, and the presence of periovular granulomatous regions). In the Sm group, there is the presence of chronic fibrous granuloma. In the group treated with HGPRT, less fibrous process occurred in the Sm group. In the group treated with the MIX of enzymes, the granulomas are partially destroyed (more destroyed when compared to the Sm + PNP group).

Table 2 shows the amount of both eggs and granulomas found in liver tissue. The number of granules was reduced by 42.1% in the liver of animals treated with HGPRT and 3.19% treated with MIX. The number of eggs was 33.22% lower in the liver of animals treated with HGPRT and 26.94% treated with MIX. PNP treatment showed no reduction in the number of granulomas or eggs.

Table 2 The amount of both eggs and granulomas found in liver tissue. Decrease in the proportion of eggs and granulomas (in %) on the 55th day after infection.

## 4. Discussion

Schistosomiasis is a disease of great importance worldwide; it is estimated that more than 256 million people are affected in 78 countries, which contributes to socioeconomic impairment. Despite the use of the medication PZQ, there are no indexes that point to a decrease in the disease for decades. Schistosomiasis continues to advance geographically, and there are no vaccines or new drugs available. Additionally, there is a decrease in sensitivity to PZQ, which has been used for the last 45 years. The magnitude of this cost can be estimated by studying the biological parameters that involves the development of resistance, such as cercariform production, infectious, reproductive success and fecundity of adult worms [6,20,21,22,23].

Developing countries are those affected by schistosomiasis and, therefore, there is no interest from the pharmaceutical and biotechnology industries in developing new drugs and vaccines for the eradication of schistosomiasis. Thus, researchers from around the world, but mainly from countries where the disease occurs, generate new knowledge to improve the health of the population at risk [24].

Once the adult worm manages to escape the host’s immunity, it is the eggs’ soluble antigens that trigger the immune response, activating antibodies, mainly IgE, that leads to the synthesis of cytokines, and the modulation of Th1 and Th2 responses, generating the effects caused by the disease [25].

The presented study proposed immunotherapy with the recombinant enzymes HGPRT, PNP and MIX of *S. mansoni* in animals, even before confirmation of egg deposition. The data obtained are important, since immunotherapy would be applied as a prophylactic approach in regions where schistosomiasis is endemic and, mainly, to understand the mechanisms of these enzymes in immature adult worms, since PZQ is not effective against them, as well as against the schistosomulae [7].

The *S. mansoni* eggs are the main cause of disease-related morbidity [25] and, therefore, a vaccine, treatment or therapy that is capable of reducing the number of eggs is essential. The results of this study show that the use of the recombinant enzymes from *S. mansoni* HGPRT, PNP and MIX significantly decreased the eggs/gram of feces, in the liver and, consequently, the number of granulomas. HGPRT caused a reduction in the number of eggs in feces and in the liver, while PNP only reduced the number of eggs in feces. Immunotherapy with these enzymes suggests that they may represent an important aspect for egg laying in females of *S. mansoni*, and these data support the idea that mainly HGPRT has a fundamental role in this process. Another study that used HGPRT as immunization and not therapy and other enzymes from the *S. mansoni* purine salvage pathway has shown that immunization with this enzyme also reduced the number of eggs in the feces. According to the results, immunization with three doses of 100 μg in 200 µL of HGPRT led to a 27% reduction in the number of eggs in feces [26]. Another study showed that immunization with 100 μg in 200 µL of enzymes from the ADK and ADSL purine rescue pathway also reduced the number of eggs per gram of feces in infected mice [17]. A recent study showed that immunization with recombinant AK and HGPRT decreased eggs/gram of feces of infected BALB/c mice [16]. Our data showed higher values when compared to the mentioned studies, with a reduction of 59–60% in the amount of eggs found in excrement in the animals immunized with total proteins found in the tegument of *S. mansoni* [16], and a reduction of 52–60% in the eggs’ amount in the feces of animals that received a non-soluble recombinant protein as an immunization expressed in the gastrodermis and tegument of adult *S. mansoni* worms [27]. Therefore, immunization and immunotherapy with enzymes from the purine salvage pathway of *S. mansoni* seem to contribute to the reduction of eggs in infected mice.

Reducing the number of eggs is essential to control the morbidity of the infection, because it helps to decrease the transmission of the disease and the formation of granulomas, which are the result of exacerbated immune and inflammatory responses against the soluble antigens of the eggs. [28].

A female of *S. mansoni* can lay around 300 eggs per day. The laying occurs in the intestinal capillary vessels of the host, then they cross into the intestinal lumen and are eliminated in the feces. About 60% of these eggs are eliminated in the feces, and the rest adhere to the intestine, liver and spleen [29]. The results of this study demonstrated that the treatment with the recombinant enzymes of *S. mansoni* HGPRT, PNP and MIX seems to cause a reduction in the parasitic load and, consequently, interfere in the lifecycle of the worm, decreasing the posture of the eggs, infection and its consequences and the PNP enzyme alone was able to generate an important reduction in the number of worms in BALB/c mice. The value obtained in the present study on the reduction of the parasite load (30.06% in the Sm + HGPRT group in Swiss mice) is superior to the value obtained for the decrease from 16.21% to 25% in the values of the parasite load after a recombinant insoluble protein that is expressed in the gastrodermis and integument of adult *S. mansoni* worms as a kind of vaccination [27]. In studies with the NDPK and HGPRT enzymes from the *S. mansoni* rescue route used in immunization in mice infected with *S. mansoni*, it was showed a slight reduction in the number of worms [16,17]. The decrease in the burden of worms after teraphy with HGPRT therapy is remarkably similar to the levels attained in additional tests using others enzymes in the immunization of mice, where worm recovery was between the 45th and the 50th day following infection. This is the case of the study of the immunization with Sm14 (Since 2012, a vaccine component has been approved for clinical studies in Brazil) and subsequent infection with 100 cercariae, which generated a reduction of 36.9–49.5% in the parasitic load [30]. The studies with a vaccine using Sm-p80 from *S. mansoni*, which is in phase 1 of the clinical trials initiated, followed by infection with 150 cercariae, showed a 46.87% reduction in the parasitic load [13,31] and a 27% reduction after immunization.

The combination of data on the decrease in the values of eggs adult worms recovered shows that using recombinant worm enzymes is an important approach to reduce the parasitic burden. The reduction of females decreases the laying of eggs and, consequently, the formation of granulomas. The data from this study show that there was a reduction in the eggs and granulomas in the liver of animals treated with HGPRT and the MIX of enzymes; that relates to the results of the treatment with PNP, where the number of eggs in feces showed an important decrease. That suggests that both enzymes can reduce the complications of the disease caused by the inflammatory reaction of the host’s immune system, which leads to the formation of the granulomas [25].

During the chronic phase of schistosomiasis, eggs are retained in the tissues. These eggs generate the granulomatous reaction, secreting proteolytic enzymes that recruit eosinophils, lymphocytes, plasmocytes and macrophages [2]. The liver is the organ most affected by this process, as the eggs are transported by the blood network and are trapped in the hepatic sinusoids, causing fibrosis [32]. The formed granuloma has, in its composition, inflammatory cells, such as eosinophils, in addition to components of the extracellular matrix, adhesion proteins, growth factors and angiogenesis. After the egg dies, the granuloma shrinks, leaving behind fibrous plaques (which includes plenty of collagen) in its place and raising portal blood pressure and portal vein diameter [25,33]. Therefore, there is portal hypertension, gastrointestinal varices, collateral circulation, ascites, and hepatomegaly. This picture characterizes hepatic fibrosis with collagen deposition [2,25]. The images of histological sections showed that the fibrosis process is more accentuated in animals that were treated with the HGPRT enzyme, as the eggs were more likely to be destroyed and the fibrous process was also accentuated, showing that HGPRT can positively interfere in the physiopathology process of schistosomiasis that severely affects the liver. The same response profile was observed in the study that tested the enzymes NDPK, ADSL and MIX in mice as immunizing agents, showing that the purine salvage pathway enzymes of *S. mansoni* interfere with the reduction of liver fibrosis [17].

The *S. mansoni* worms have a marked sexual dimorphism and, consequently, the expression of proteins in the sexes is different. A study has shown that *S*. *mansoni* females have a preferential expression of APRT in adult and mature female gonads [34]. The APRT enzyme is present in the pathway of converting adenosine in nucleotides in *S. mansoni* to purines. This enzyme catalyzes the condensation reaction between adenine and PRPP (5-phosphoribosylpyrophosphate) to produce AMP and PPi. Kinetic experiments using the heterologous enzyme of a *S. mansoni* APRT isoform indicated that this enzyme is catalytically active [34]. A study using in-situ hybridization indicated that the transcripts of this enzyme are concentrated in the posterior ovary region of adult females [34]. Therefore, the study of enzymes present in the purine metabolism of *S. mansoni* and the analysis of the proportion of worms recovered (couples, males and females) could contribute to the understanding of the infection modulation, and, consequently, to the interruption of the parasite lifecycle. The results of the present study showed a reduction in males in the groups treated with HGPRT, PNP and the MIX of enzymes and females in the groups treated with HGPRT and PNP, when compared to couples.

It is well known that eosinophils act as effector cells to prevent worm infection, but their exact role is still debated. To act at sites of inflammation/infection, eosinophils are recruited, and that response may contribute to the reduction of circulating eosinophils [35,36,37]. As the results showed, eosinophils increased in number in PCL and PB, being this more important in PLC from infected animals, when compared to the control group.

In the post-inflammatory fibrotic process, eosinophils can stimulate collagen synthesis through releasing cytokines and granules that affect with the characteristics of fibroblasts which makes these cells important for controlling the pathophysiology of the disease. Our results showed that the number of eosinophils in the blood and PCL of animals treated with HGPRT is similar to the infected group because of the infection stimulus, it hints at a distinctive eosinophilia in this kind of helminthiasis. On the other hand, the Sm + PNP and Sm + MIX group presented a negative regulation of this eosinophilia, but also showed a decreased of granulomas and eggs in the hepatic tissue. The explanation for this would be a process of migration of eosinophils to the organ, assisting in the removal of eggs and, as a result, to the reduction in the number of granulomas.

A work reported the acquisition of resistance in individuals from endemic areas and reinfection by IgE antibody-mediated reactions during schistosomiasis mansoni [2]. Significant levels of IgE antibody production are seen in outcomes on therapy with recombinant GHPRT, PNP, and MIX. IgE antibodies activate skin mast cells and blood basophils, that are recognized to to play a direct effector role in immunization against schistosomiasis and be implicated in acute hypersensitivity reactions. This type of antibodies has also been associated with reduced parasite burden and decreased parasite fertility in experimental models [38], which was also observed in the immunization with the recombinant enzymes MIX in our study. The data showed that moderate levels of IgE were produced in the infected and treated groups with the recombinant enzymes of *S. mansoni*. In addition, the plasma of these animals also recognized the total proteins of *S. mansoni* (PTS). IgE antibodies can activate mast cells and basophils in the skin and blood, respectively, generating an immediate hypersensitivity. Furthermore, studies showed that the production of this class of antibody is associated with reduced parasitic load and decreased fertility of adult worms in different models, which corroborates all the data obtained in the present study [38].

The cytokine IL-4 is important to initiate the type Th2 response, which is important in schistosomiasis physiopathology, and is well established to be essential for altering the IgE antibody type [39]. If there is a depletion of this response, it can cause tissue damage and even the death of the patient, due to the pro-inflammatory action Th1. It is, therefore, a response that can act in favor of the patient, protecting and minimizing the damage caused by the parasite [40]. IL-4 is also responsible for the formation of fibrosis, as it activates fibroblasts to produce collagen [25]. Our data demonstrate that therapy with HGPRT and the MIX of the enzymes significantly increased IL-4 concentrations in the plasma of Balb/c mice.

The enzyme HGPRT could reduce the physiopathological aspects that lead to related morbidity, such as liver fibrosis and positive modulation for IL-4 and IgE expression, while the enzyme PNP showed effect on the worm itself, reducing the parasitic load and number of eggs. Therefore, both enzymes cause an impact on some important aspects of schistosomiasis and could be extremely promising as candidates in the search for therapies and vaccines against this important disease.

## Data Availability

Data supporting the reported results can be found in the repository of the institution where the study was carried out: https://repositorio.ufscar.br/handle/ufscar/10614 (assessed on 20 January 2023) and https://repositorio.ufscar.br/handle/ufscar/13333 (assessed on 20 January 2023).

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
