# Peer review of "HGPRT and PNP: Recombinant Enzymes from Schistosoma mansoni and Their Role in Immunotherapy during Experimental Murine Schistosomiasis"

_pathogens, 2023, doi:10.3390/pathogens12040527_

Round 1
Reviewer 1 Report
The manuscript describes the therapy with recombinant enzymes from Schistosoma mansoni HGPRT, PNP and the MIX of both enzymes in experimental murine schistosomiasis. This subject is important since schistosomiasis is a global problem and current medication has some limitations. The manuscript is well organized and well written. However, certain points need to be revised.
1. Why in the experiment with female Swiss mice only HGPRT treatment was used?
2- Please, correct the word "Rrecovered" in the Fig 2B.
3- The caption of figure 3 is incorrect.
Author Response
Dear reviewer, thank you for your reviews and suggestions. Changes are highlighted in red in the text.
1. Why in the experiment with female Swiss mice only HGPRT treatment was used?
The study using only HGPRT as a treatment was previously carried out in our laboratory. As the results proved to be very promising, a new project was established with one more enzyme from the purine salvage pathway, PNP, now using the Balb/c strain and with further analyses.
2- Please, correct the word "Rrecovered" in the Fig 2B.
The word has been corrected.
3- The caption of figure 3 is incorrect
The caption has been corrected.
Reviewer 2 Report
The present study evaluated the effects of the recombinant enzymes of S. mansoni Hypoxanthine-Guanine Phosphoribosyltransferase (HGPRT), Purine Nucleoside Phosphorylase (PNP) and the MIX of both enzymes in the immunotherapy of schistosomiasis in murine model. These enzymes are part of the purine salvage pathway, the only metabolic pathway present in the parasite for this purpose, being essential for the synthesis of DNA and RNA. Female mice of Swiss and BALB/c strains were infected with cercariae and treated, intraperitoneally, with three doses of 100 µg of enzymes. After the immunotherapy, the eggs and adult worms were counted in the feces, the number of eosinophils from the fluid in the peritoneal cavity and peripheral blood was observed, and the quantification of the cytokine IL-4 and the production of antibodies IgE was analyzed. The evaluation of the number of granulomas and collagen deposition via histological slides of the liver was performed. The results demonstrate that immunotherapy with the enzyme HGPRT seems to stimulate the production of IL-4 and promoted a significant reduction of granulomas in the liver in treated animals. The treatment with the enzyme PNP and the MIX was able to reduce the number of worms in the liver and in the mesenteric vessels of the intestine, to reduce the number of eggs in the feces and to negatively modulate the number of eosinophils. Therefore, immunotherapy with the recombinant enzymes of S. mansoni HGPRT and PNP might contribute to the control and reduction of the pathophysiological aspects of schistosomiasis, helping to decrease the morbidity associated with the infection in murine model.
This is topic relevant because although the studies that they present are preliminary, trying to find a vaccine for the effect of the recombinant enzymes over of S. mansoni Hypoxanthine-guanineosphoribosyltransferase (HGPRT), Purine Nucleoside Phosphorylase (PNP) and the MIX are important of both enzymes in part of the purine salvage pathway, the only metabolic pathway the immunotherapy of schistosomiasis in murine model. These enzymes are present in the parasite for this purpose. The controls that should be considered is the use of this vaccine including different concentrations of the vaccine in order to find the optimal concentration of this vaccine.
Author Response
Dear reviewer, thank you for your reviews and suggestions.
Reviewer 3 Report
Summary:
Fragelli et al. report on the results of their study focussed on potential targets for immunotherapy against schistosomiasis. The study further examine the role of the recombinant Schistosoma mansoni enzymes (HGPT and PNP) by targeting the stages of the parasite development less vulnerable to the recommended drug, praziquantel.
In my opinion, the study was well-designed with appropriate controls and the conclusions generally consistent with the data presented. However, I have comments, questions and suggestions below to further improve the manuscript.
Comments:
Line 99: ‘Materials and methods’
Line 101: Please, provide more details on the cloning and expression process for the enzymes. For example, which expression vector was used? What was the original source of Schistosoma mansoni used – wild or lab strain? If available, the GenBank accession number of the proteins. Were these enzymes designed and expressed to be functional or not?
Line 126: What is the rationale for the 100 µg dose? Was this from preliminary studies or some estimates? This should be included for clarity of the method.
Lines 146-147: Here it says eggs were counted on 38th and 54th days but in line 141, it says egg counting was on day 54. Please, correct that.
Line 178: It seems from the text under this section that other cytokines beside IL-4 were measured. Can the title be modified to reflect this fact?
Lines 186-188: There is need to clarify some of this information. What is the role of the sample (adult worms) as described here in the measurement of cytokines and IgE?
Line 239: Change 2C to 1C
Line 245: In Figure 1C & 1D, the HGPRT (green bar). Where did this come from? My interpretation is that this was a group not infected with S. mansoni but treated with HGPRT. However, this was not included in the experimental groups described earlier in Table 1. Secondly, why is the data shown differently for Swiss vs Balb/c mice. For example Sm+HGPRT is shown in Figure 1A for Swiss mice but in Figure 2A, there are more groups shown. Is that how the experiment was done or there is a rationale for why the data is shown that way. Please, clarify in the text. Otherwise it is not clear.
Lines 276-279: Figure 3 legend. There is inconsistency between this and the text in lines lines 269-275. Please, reconcile this information.
Line 316: While most current control programs still emphasize mass treatment with praziquantel, there are efforts based on control of intermediate hosts as well. See Sokolow SH, Wood CL, Jones IJ, Swartz SJ, Lopez M, Hsieh MH, et al. (2016) Global Assessment of Schistosomiasis Control Over the Past Century Shows Targeting the Snail Intermediate Host Works Best. PLoS Negl Trop Dis 10(7): e0004794. https://doi.org/10.1371/journal.pntd.0004794
Lines 347-355: Apparently there are a couple of immunization studies using the HGPRT enzyme. It would be great if you can include in your studies why it was necessary to use this enzyme again. How different is your studies from the referenced ones with respect to HGPRT? Perhaps there was another aspect of it that you aimed to address with this study…
The study is within an area in need of further research to identify solutions for lasting treatment or elimination of schistosomiasis. However, the specific approach of this study is not entirely novel. Also, I think that the manuscript from benefit from revision for clarity of the methods and presentation of results.
Author Response
Dear reviewer, thank you for your reviews and suggestions. Changes are highlighted in red in the text.
Line 99: ‘Materials and methods’
Corrections have been made.
Line 101: Please, provide more details on the cloning and expression process for the enzymes. For example, which expression vector was used? What was the original source of Schistosoma mansoni used – wild or lab strain? If available, the GenBank accession number of the proteins. Were these enzymes designed and expressed to be functional or not?
Corrections have been made. And yes, enzymes were designed to be functional.
Line 126: What is the rationale for the 100 µg dose? Was this from preliminary studies or some estimates? This should be included for clarity of the method.
The dose was established by the standardization carried out by the research group, which was based on the bibliography
Lines 146-147: Here it says eggs were counted on 38th and 54th days but in line 141, it says egg counting was on day 54. Please, correct that.
Corrections have been made.
Line 178: It seems from the text under this section that other cytokines beside IL-4 were measured. Can the title be modified to reflect this fact?
Corrections have been made.
Lines 186-188: There is need to clarify some of this information. What is the role of the sample (adult worms) as described here in the measurement of cytokines and IgE?
Plate sensitization with PTS for IgE measurement guarantees that the positive reaction for HGPRT and PNP is specific for these two proteins.
Line 239: Change 2C to 1C
Corrections have been made.
Line 245: In Figure 1C & 1D, the HGPRT (green bar). Where did this come from? My interpretation is that this was a group not infected with S. mansoni but treated with HGPRT. However, this was not included in the experimental groups described earlier in Table 1. Secondly, why is the data shown differently for Swiss vs Balb/c mice. For example Sm+HGPRT is shown in Figure 1A for Swiss mice but in Figure 2A, there are more groups shown. Is that how the experiment was done or there is a rationale for why the data is shown that way. Please, clarify in the text. Otherwise it is not clear.
The study using only HGPRT as a treatment was previously carried out in our laboratory. As the results proved to be very promising, a new project was established with one more enzyme from the purine salvage pathway, PNP, now using the Balb/c strain and with further analyses.
Lines 276-279: Figure 3 legend. There is inconsistency between this and the text in lines lines 269-275. Please, reconcile this information.
Corrections have been made.
Line 316: While most current control programs still emphasize mass treatment with praziquantel, there are efforts based on control of intermediate hosts as well. See Sokolow SH, Wood CL, Jones IJ, Swartz SJ, Lopez M, Hsieh MH, et al. (2016) Global Assessment of Schistosomiasis Control Over the Past Century Shows Targeting the Snail Intermediate Host Works Best. PLoS Negl Trop Dis 10(7): e0004794. https://doi.org/10.1371/journal.pntd.0004794
Corrections have been made.
Lines 347-355: Apparently there are a couple of immunization studies using the HGPRT enzyme. It would be great if you can include in your studies why it was necessary to use this enzyme again. How different is your studies from the referenced ones with respect to HGPRT? Perhaps there was another aspect of it that you aimed to address with this study…
Previous studies with HGPRT are for immunization. The present study sought to evaluate not the immunization, but the treatment of schistosomiasis, which is the differential.